# Generation of a *Gluconobacter oxydans* knockout collection for improved extraction of rare earth elements

Alexa M. Schmitz[1], Brooke Pian [1], Sean Medin[1], Matthew C. Reid[2], Mingming Wu [1], Esteban Gazel [3] & Buz Barstow [1✉]

Bioleaching of rare earth elements (REEs), using microorganisms such as *Gluconobacter oxydans*, offers a sustainable alternative to environmentally harmful thermochemical extraction, but is currently not very efficient. Here, we generate a whole-genome knockout collection of single-gene transposon disruption mutants for *G. oxydans* B58, to identify genes affecting the efficacy of REE bioleaching. We find 304 genes whose disruption alters the production of acidic biolixiviant. Disruption of genes underlying synthesis of the cofactor pyrroloquinoline quinone (PQQ) and the PQQ-dependent membrane-bound glucose dehydrogenase nearly eliminates bioleaching. Disruption of phosphate-specific transport system genes enhances bioleaching by up to 18%. Our results provide a comprehensive roadmap for engineering the genome of *G. oxydans* to further increase its bioleaching efficiency.

[1] Department of Biological and Environmental Engineering, Cornell University, Ithaca, NY 14853, USA. [2] School of Civil and Environmental Engineering, Cornell University, Ithaca, NY 14853, USA. [3] Department of Earth and Atmospheric Sciences, Cornell University, Ithaca, NY 14853, USA.
✉email: bmb35@cornell.edu

Rare earth elements (REEs) are essential for the manufacturing of modern electronics[1–3] and sustainable energy technologies, including electric motors and wind turbine generators[4], solid-state lighting[5], battery anodes[6], high-temperature superconductors[7], and high-strength lightweight alloys[8,9]. All of these applications place increasing demands on the global REE supply chain[10]. As the world demand for sustainable energy grows[11], finding a reliable and sustainable source of REEs is critical.

Current methods for refining REEs often involve harsh chemicals, high temperatures, and high pressures and generate a considerable amount of toxic waste[12]. These processes give sustainable energy technologies reliant on REEs a high environmental and carbon footprint. As a consequence, due to its high environmental standards, the United States has no capacity to produce purified REEs[13,14].

There is growing interest in biological methods to supplement, if not completely replace, traditional REE extraction and purification methods[15–19]. Biological extraction (bioleaching) is already used to extract 5% of the world's gold[20,21] and ≈15% of the world's copper supply[20,21] (in fact, Cu biomining in Chile alone accounts for 10% of the world's Cu supply[22,23]).

The performance of REE bioleaching lags behind thermochemical processes. For example, while thermochemical methods have 89–98% REE extraction efficiency from monazite ore[24,25], Aspergillus species can only achieve ≈3–5%[16]. The acid-producing microbe Gluconobacter oxydans B58 can recover ≈50% of REEs from FCC catalysts[26]. However, techno-economic analysis indicates that even this extraction efficiency is still not high enough for commercial application[19].

Recent efforts to improve bioleaching have focused exclusively on process optimization[27]. To our knowledge, no genetic approaches have yet been taken for any bioleaching microbe[28]. With recent advances in tools for reading and writing genomes, genetic engineering is an attractive solution for enhancing bioleaching. However, applying these tools to non-model microorganisms such as G. oxydans can be a significant challenge[9]. While there have been some promising advances in editing the G. oxydans genome[29–34], we do not yet know where to edit.

In the presence of glucose, G. oxydans secretes a biolixiviant rich in gluconic acid[26]. This is produced by periplasmic glucose oxidation by the pyrroloquinoline quinone (PQQ)-dependent membrane-bound glucose dehydrogenase (mGDH)[35]. The final pH of the biolixiviant is a major factor in REE bioleaching[26]. However, gluconic acid alone fails to explain bioleaching by G. oxydans: pure gluconic acid is far less effective at bioleaching than the biolixiviant produced by G. oxydans[26]. This means that even the most successful efforts to upregulate mGDH activity and gluconic acid production are unlikely to take full advantage of G. oxydans' biolixiviant production capabilities.

In this work, we characterize the genome of G. oxydans and identify a comprehensive set of genes underlying its bioleaching capabilities through the construction of a carefully curated whole-genome knockout collection of single-gene transposon disruption mutants using Knockout Sudoku[36,37] (Fig. 1). As the final pH of the biolixiviant is a good predictor of bioleaching efficiency[26]; we use acidification as a proxy for bioleaching potential and screen the collection for mutants that differ in their ability to produce acidic biolixiviant (Figs. 2 and 3). Finally, we demonstrate that a single gene disruption - only one of several potential enhancement strategies—can already significantly improve G. oxydans' bioleaching capabilities (Fig. 4).

## Results

### Development of a knockout collection for G. oxydans. 
We built a saturating coverage transposon insertion mutant collection for G. oxydans B58 and cataloged and condensed it with the Knockout Sudoku combinatorial pooling method[36,37] (Fig. 1). We sequenced

the G. oxydans B58 genome and identified 3283 open reading frames (Fig. 1A and Supplementary Data 1; see "Methods" section). Following the recommendation of Monte Carlo simulations, we collected 49,256 transposon insertion mutants (the progenitor collection; PC) to ensure saturating coverage of the G. oxydans B58 genome (Fig. 1B; see "Methods" section).

Progenitor collection sequencing results indicate that we were able to generate at least one disruption mutant for almost every nonessential gene in the G. oxydans genome. In total, we identified disruption strains for 2733 genes out of the 3283 genes in the G. oxydans B58 genome. Since every predicted gene contains at least seven AT or TA transposon insertion sites (Supplementary Data 1), the remaining 550 non-disrupted genes are likely to be essential. A Fisher's Exact Test for gene ontology (GO) enrichment representing 268 of the non-disrupted genes demonstrated significant enrichment ($p < 0.05$) in several essential ontologies, with the greatest enrichment in those relating to the ribosome and translation (Fig. 1C and Supplementary Data 2A).

The progenitor collection catalog was used to create a condensed G. oxydans disruption collection with at least one representative per nonessential gene. Forty-seven progenitor strains were verified by Sanger sequencing prior to condensing, of which 43 (92%) were confirmed to have the predicted transposon coordinate (Supplementary Data 3A). We selected one mutant for all 2733 disrupted genes, a second mutant for 2354 genes, and a third mutant for 50 genes where mutant location information was poor. All mutants were transferred for isolation of single colonies, and 2–10 colonies per mutant were picked, depending on the predicted number of cross-contaminating disruption strains in the originating well. This condensed collection contains 17,706 mutants in 185 96-well plates (Supplementary Data 4).

The condensed collection catalog was validated by a second round of combinatorial pooling and sequencing. Of the 17,706 wells in the condensed collection, we were able to confirm the identity of 15,257 (Supplementary Data 4). We confirmed 25 of these wells by Sanger sequencing, and 100% had the predicted transposon coordinates (Supplementary Data 3B). Among these wells, we were able to verify the identity and location of 4419 independent transposon insertion sites, representing a disruption mutant for 2556 unique genes (Fig. 1D). One thousand five hundred and eighty seven genes are represented by more than one disruption, and 3317 of all disruptions occur in the first half of the gene (Supplementary Data 4).

### Genome-wide screening discovers genes linked to acid production. 
We screened the G. oxydans B58 whole genome knockout collection for disruption mutants with differential acidification capability (Fig. 2). We used the colorimetric pH-sensitive dye thymol blue (TB) to screen for changes in final biolixiviant pH (Fig. 2A and S1) and bromophenol blue (BPB) to screen for changes in the rate of acidification (Fig. 2B).

In total, we noted 304 genes that apparently controlled acidification (Fig. 2C and Supplementary Data 5). The TB screen discovered 282 genes whose disruption led to a differential change in biolixiviant acidity (Fig. 2C and Supplementary Data 5). Forty-seven mutants produced a more acidic biolixiviant, while 235 produced a less acidic biolixiviant (Fig. 2C and Supplementary Data 5). The BPB screen identified 82 gene disruptions with differential rates of acidification: 49 with a faster rate and 33 with a slower rate. Sixty mutants were identified by both screens (Fig. 2C and Supplementary Data 5).

Overall, we identified 165 genes that significantly ($p < 0.05$) changed the final biolixiviant pH, rate of acidification, or both but did not change the growth rate (Fig. 2C and Supplementary Data 6). We re-arrayed disruption strains with differential acidification into new 96-well plates alongside proxy wild-type (pWT) strains

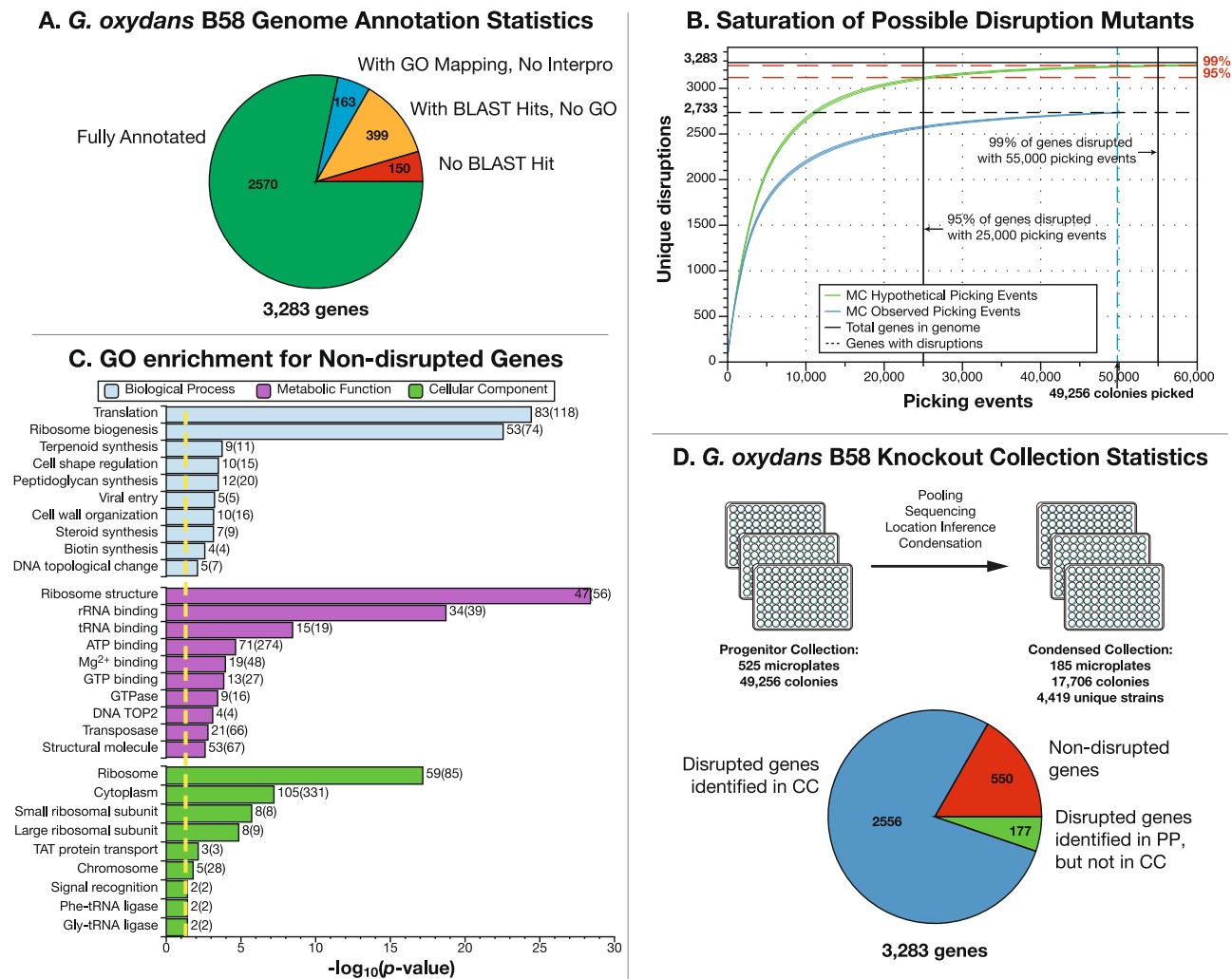

**Fig. 1 Knockout Sudoku was used to curate a saturating coverage transposon insertion mutant collection for *Gluconobacter oxydans* B58. A** The *G. oxydans* B58 genome contains 3283 genes. Two thousand five hundred and seventy genes were fully annotated with a BLAST hit, InterPro ID, and gene ontology (GO) group. An additional 163 genes had an annotation and GO group but lacked an InterPro ID; 399 retrieved only a BLAST hit, and 150 were unable to be assigned any annotation. **B** A Monte Carlo (MC) estimate (green curve) of the number of genes represented by at least one mutant as a function of the number of mutants collected demonstrated that picking 25,000 mutants would yield at least one disruption for 95% of genes, while picking 50,000 mutants would yield at least one disruption for 99% of genes. In total, we picked 49,256 single-gene disruption mutants and located at least one disruption for 2733 genes. A Monte Carlo simulation (blue curve) of picking with random drawing from the sequenced progenitor collection (PC) without replacements demonstrates that the genome coverage was truly saturated. The center of each curve is the mean value of the unique gene disruption count from 1000 simulations, while the upper and lower parts of each curve represent two standard deviations around this mean. **C** A one-sided Fisher's exact test for gene ontology enrichment among the non-disrupted (putatively essential) genes revealed significant enrichment ($p < 0.05$, yellow line) of genes involved in translation and other ribosome-related functions. **D** The curated condensed collection (CC) contains 17,706 isolated colonies across 185 plates. High-throughput sequencing of the CC confirmed the location for 4419 unique disruption strains, representing disruptions in 2556 genes. Hundred and seventy-seven genes located in the PC were not located in the CC. No disruption mutant was detected in 550 genes.

(see "Methods" section) that have a transposon insertion in an intergenic region and show nondifferential growth or biolixiviant production (Supplementary Fig. 2). The new collection was re-assayed with TB and BPB assays, and the strength and significance of each result were determined by comparison with pWT through a Bonferroni-corrected *t*-test (see "Methods" section). Mutants that cause the 25 largest reductions and 50 largest increases in endpoint acidity yet do not affect the growth rate are shown in Fig. 2D. A full set of mutants that cause significant changes in acidification are listed in Supplementary Data 6.

Thirty-one mutants that caused significant changes in the acidification rate without changing the growth rate are shown in Fig. 2E and Supplementary Data 6F. However, 14 of the faster strains, including δGO_868, a disruption of a LacI-type

transcriptional repressor, which was the fastest strain, produced a less acidic biolixiviant than the wild-type, indicating that targeting these genes for engineering a faster acidifier would likely be at the expense of a more acidic biolixiviant. None of the strains with a faster rate of acidification also created a more acidic biolixiviant. This result suggests that multiple genetic engineering interventions will be needed to construct a strain of *G. oxydans* that simultaneously produces a more acidic biolixiviant than the wild-type at a faster initial rate.

**Pi transport and PQQ synthesis are main controllers of acidification.** We used Fisher's exact test to determine which biological processes, metabolic functions, and cellular components are enriched

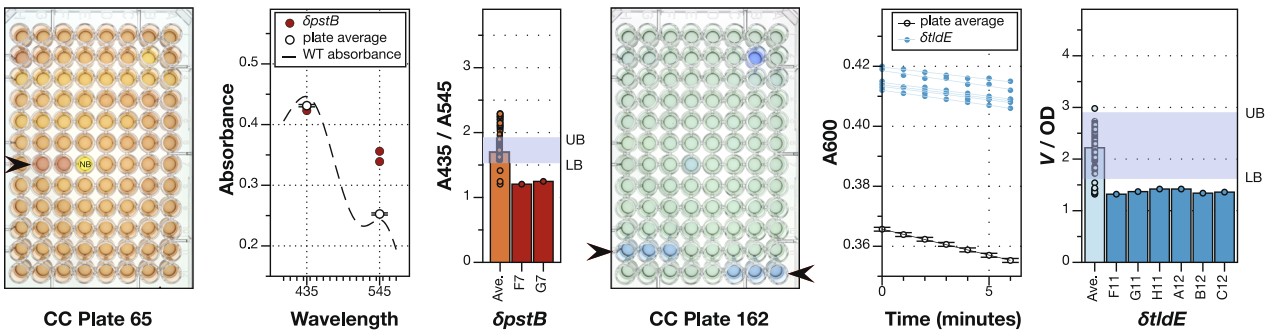

**A. TB Screen Finds Knockout of Gene Involved in Phosphate Transport Reduces Final Acidity**

**B. BPB Screen Finds Knockout of Gene Likely Involved in PQQ Synthesis Slows Acidification Rate**

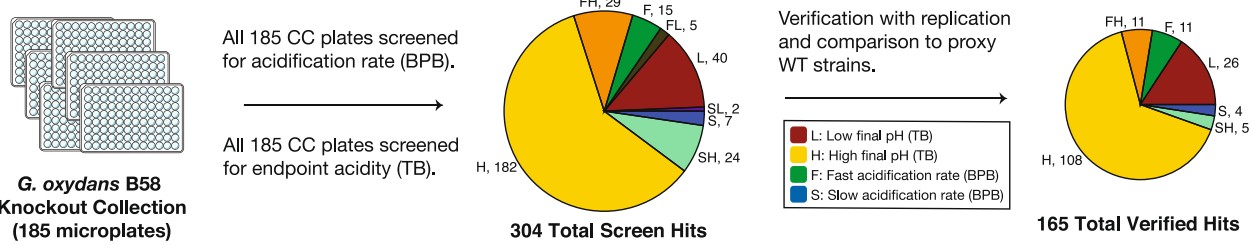

**C. Work flow for identifying comprehensive set of genes underlying acidification**

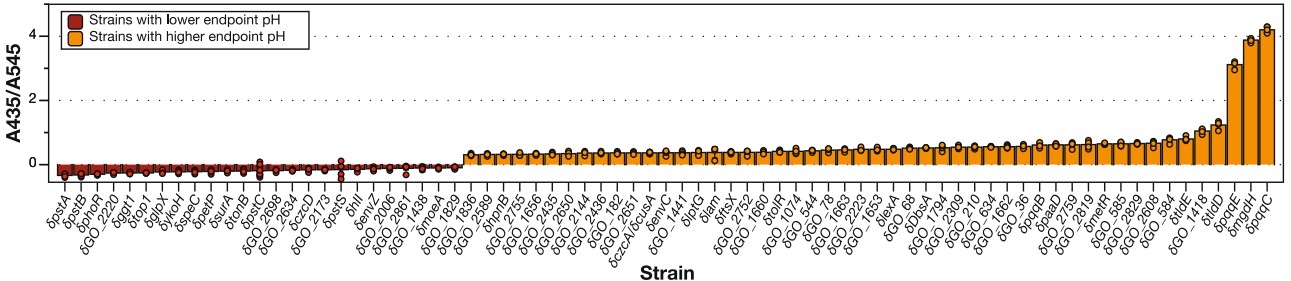

**D. 25 Biggest Reductions and 50 Biggest Increases in Biolixiviant Acidity**

**E. Most Significant Changes in Acidification Rate**

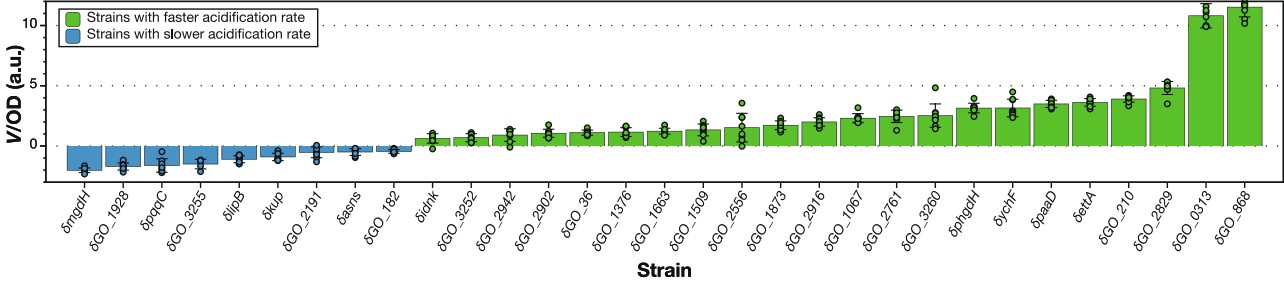

among the gene disruptions that significantly changed biolixiviant acidification (Fig. 3). Among the disrupted genes that led to a stronger acidity, the most significant enrichment for all three GO categories involves the phosphate-specific transport system, represented by *pstA*, *pstB*, *pstC*, *pstS*, and *phoR* (Fig. 3A). Other enriched ontologies include those related to phosphate signaling and binding.

Among the disrupted genes that led to weaker acidity, several enriched GO terms are related to the synthesis or use of the redox cofactor PQQ, represented by *pqqB*, *pqqC*, *pqqE*, *tldD*, and *mgdh* (Fig. 3B). Other enriched ontologies include those related to carbohydrate metabolism, represented by five enzymes related to fructose metabolism: Fructokinase (*GO_1072*), GDP-mannose 4,6-dehydratase (*GO_1441*), Mannose-1-phosphate guanylyl transferase/Mannose-6-phosphate isomerase (*GO_182*),

UTP--glucose-1-phosphate uridylyltransferase /Phosphoman-nomutase (*GO_1923*), and UDP-glucose 4-epimerase (*GO_36*) (Supplementary Data 2C).

Acidification rate is controlled by carbohydrate metabolism and respiration. Disruptions in the pentose phosphate pathway and related metabolic processes increased the acidification rate (Fig. 3C). Meanwhile, disruptions of the electron transport pathway components are the most significantly enriched group of mutants that decreased the acidification rate (Fig. 3D).

**Dye assay results are validated by direct pH measurements**. We selected 22 strains from those with the greatest increase or decrease in acidification (Fig. 2D, E) for further testing. Dye pH measurements were validated by direct pH measurements. Eleven

**Fig. 2 High-throughput pH screens of the *G. oxydans* whole genome knockout collection were used to identify genes that control REE bioleaching. A** Thymol blue (TB) was used to measure the endpoint acidity of biolixiviant produced by each well of the condensed collection. The ratio of TB absorbance (A) at 435 and 545 nm is linearly related to pH between 2 and 3.4 (Supplementary Fig. 1). CC plate 65 contains biolixiviant produced by δ*pstB* strain in wells F7 and G7 (arrowhead), whose absorbance at 435 and 545 nm are shown (middle panel, red dots), along with the average absorbance of all occupied wells on the plate ($n = 94$) (white dot; error bars are SEM). The dashed line represents a typical absorbance spectrum for WT-produced biolixiviant. The A435/A545 ratio for these two wells (right panel, red bars and dots) compared with the mean ratio of the plate (orange bar; orange dots show individual data points) is well below the lower bound (LB) for the plate, indicating that δ*pstB* produces a much more acidic biolixiviant than the average strain. **B** Bromophenol blue (BPB) was used to measure rate of change in pH at the onset of glucose conversion to organic acids. Rate was measured over a 6 min period within five minutes of adding bacteria to a glucose and BPB solution. Condensed collection (CC) plate 162 contains the δ*tldE* strain in wells F11–C12 (arrowheads), whose changes in absorbance over time are graphed along with the average for that plate (middle panel, $n = 94$). A comparison of the normalized rate over OD for each well (right panel, dark blue bars and dots) versus the plate mean (light blue bar; dots show individual data points) shows how V/OD for these wells was below the lower bound for CC plate 162. **C** All 185 plates of the CC were screened for acidification using TB and BPB assays. Hits from both screens were verified in comparison with proxy WT strains. In total, 176 disruption strains were shown to significantly contribute to acidification. **D**, **E** Comparisons with proxy WT strains were made by a two-tailed *t*-test with a Bonferroni-corrected alpha ($\alpha = 0.05/N$ where $N$ is the number of comparisons). Bars represent mean values and dots represent individual data points. Error bars represent standard deviation. **D** The 25 largest significant reductions in biolixiviant pH and 50 largest significant increases in biolixiviant pH. $N = 120$ or $N = 242$ for comparisons with pWT set A or set B, respectively. (The full data set and number of biological replicates for each strain can be found in Supplementary Data 6E). **E** All significant changes in acidification rate, $N = 60$ (The full data set and number of biological replicates for each strain can be found in Supplementary Data 6F).

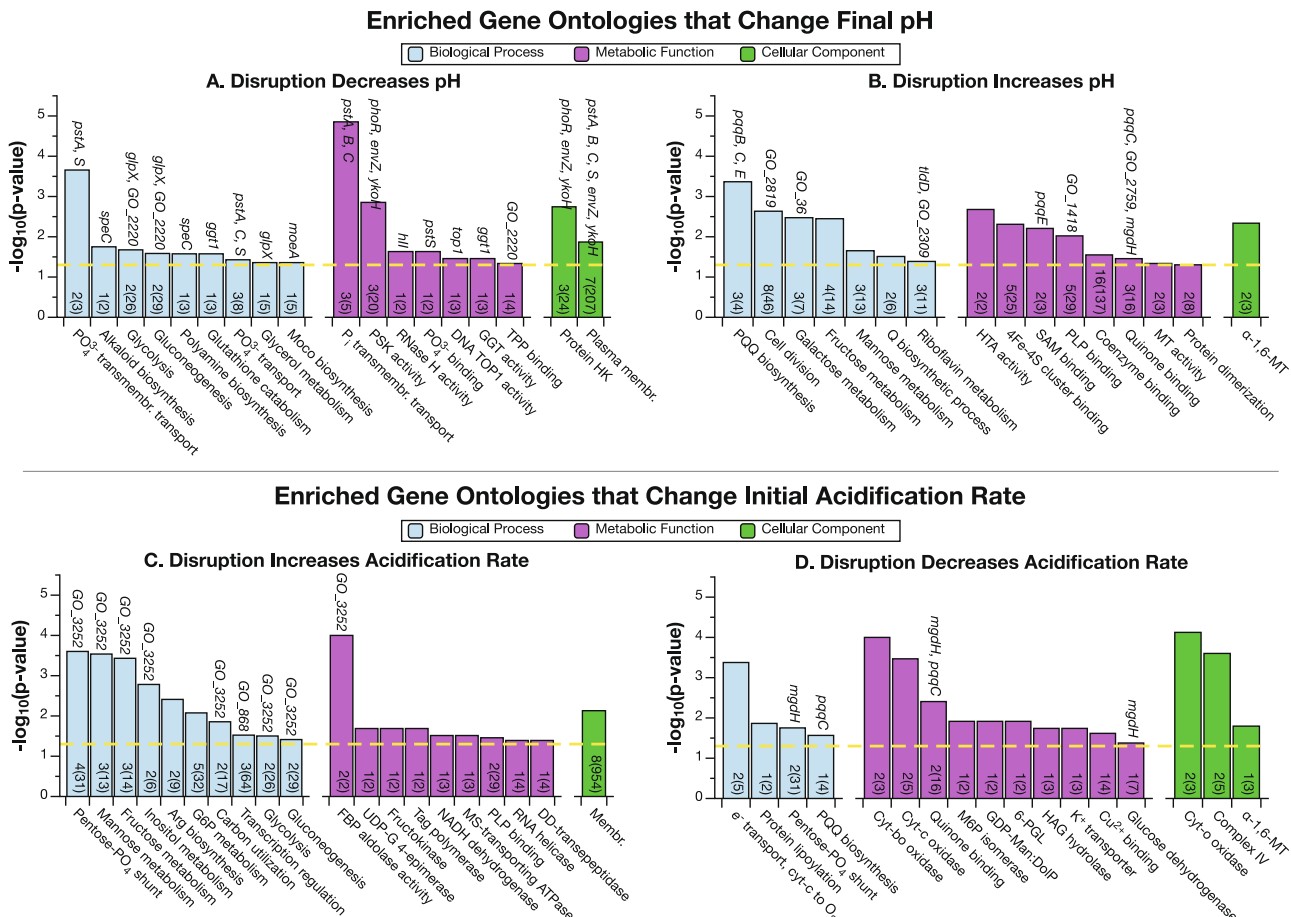

**Fig. 3 Genes involved in phosphate signaling, carbohydrate metabolism and PQQ synthesis were significantly overrepresented in the significant hits from high-throughput screens of acidification by *G. oxydans*.** A one-sided Fisher's Exact Test was used to test for gene ontology enrichment ($p < 0.05$, yellow dashed line). Numbers at the base of bars are how many genes from the significant hits are from that gene ontology (GO), out of the total in the genome (in parentheses). Genes selected for further analysis of endpoint pH and bioleaching (Fig. 4) that contribute to an enriched GO are listed above the bars. **A**, **B** Enriched GO terms among genes that decreased and increased the end point pH. **C**, **D** Enriched GO terms among genes that increased and decreased the initial acidification rate. FBP fructose-bisphosphate, GDP-Man:DolP dolichyl-phosphate beta-D-mannosyltransferase, GGT glutathione hydrolase, G6P glucose 6-phosphate, HTA homoserine O-acetyltransferase, DD-transepeptidase D-Ala-D-Ala carboxypeptidase, HAG hydroxyacylglutathione, Membr membrane, Moco Mo-molybdopterin cofactor, MS monosaccharide, MT mannosyltransferase, M6P mannose-6-phosphate, Pi inorganic phosphate, PLP pyridoxal phosphate, PQQ pyrroloquinoline quinone, PSK phosphorelay sensor kinase, Q queuosine, RNase H DNA–RNA hybrid ribonuclease, SAM S-adenosyl-L-methionine, TPP thiamine pyrophosphate, TOP1 topoisomerase type 1, HK histidine kinase, UDP-G uracil-diphosphate glucose, 6-PGL 6-phosphogluconolactonase.

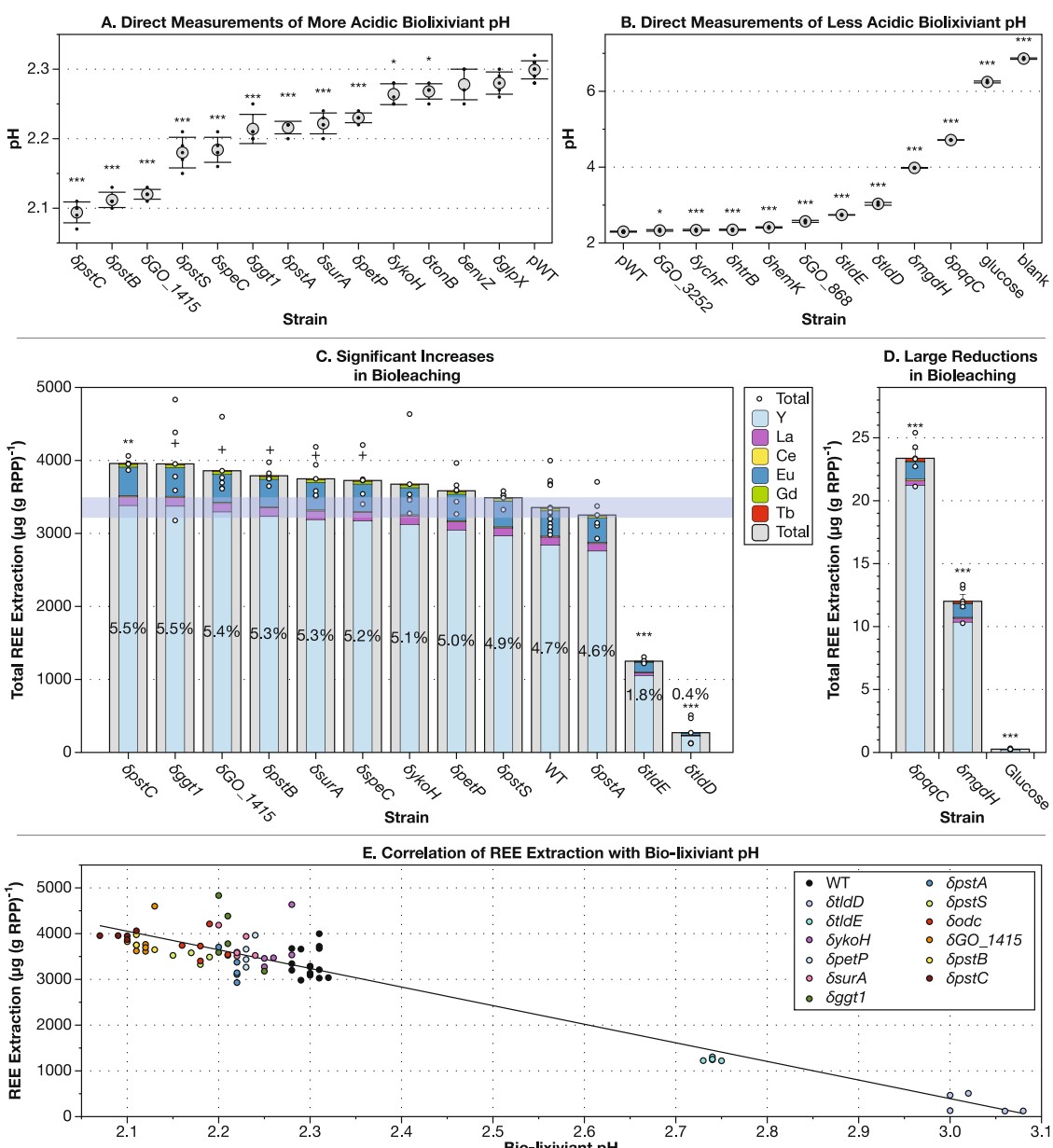

**Fig. 4 Increased acidification strains of *G. oxydans* B58 are able to increase rare earth extraction from retorted phosphor powder (RPP). A**, **B** A subset of 22 disruption strains was tested for acidification with direct pH measurement. Gray circles represent the mean pH for each strain and comparisons were made with wild type by two-tailed *t*-test with Bonferonni correction (*N* = 22). pH measurements significantly different from pWT are labeled with asterisks: *\*p* < 0.05/22; *\*\*p* < 0.01/22; *\*\*\*p* < 0.001/22 (*n* = 5 biological replicates, df = 18). Error bars represent standard deviation. Individual data points are shown as black dots. **C**, **D** Ten disruption strains with the lowest final biolixiviant pH and four with the highest were tested for RPP bioleaching capabilities. Outer gray bars represent the mean total REE extracted for each biolixiviant. Inner multicolored bars represent fractional contributions of each REE. Error bars represent standard error around the mean total REE extracted. Percentages are total REE extraction efficiency (based on previously published REE amounts in the RPP[26]). Individual data points for total REE extracted are shown as white dots. **C** Comparison with a two-tailed *t*-test between each mutant and pWT demonstrated that eight strains were significantly better or worse at bioleaching total REEs (+*p* < 0.05; *n* = 5 biological replicates, df = 18). With a Bonferroni correction (*N* = 12), only one was significantly better (*\*\*p* = 5.30 × 10⁻⁴ for δpstC), but two of the higher pH biolixiviants that extracted detectable REEs were significantly attenuated in bioleaching capability (*\*\*\*p* = 1.71 × 10⁻¹¹ and 6.32 × 10⁻¹⁴ for δtldE and δtldD, respectively). Light purple shading represents the mean and standard deviation for total REE bioleached by pWT biolixiviant. **D** Disruption mutants for *mgdh* and *pqqC* are only able to extract less than 1% of the REE that wild-type *G. oxydans* can but still extract significantly more REE than glucose alone when measured at a lesser dilution (two-tailed *t*-test with Bonferroni correction, *N* = 2: *\*\*\*p* = 9.71 × 10⁻¹⁰ and 2.47 × 10⁻⁸ for δpqqC and δmgdh, respectively, *n* = 5 biological replicates). **E** Total REE extraction is plotted versus pH for each replicate for each strain, demonstrating how the two are linearly related. Each data point is color coded by strain, as indicated in the plot legend.

of the 13 strains that produced significantly lower acidity biolixiviants in TB assays did the same in direct pH measurements (Fig. 4A). The most acidic biolixiviant was produced by a disruption in the phosphate transport gene δpstC at pH 2.09. In fact, 4 of the 11 mutant strains that produced a more acidic

biolixiviant were disrupted in genes involved in the phosphate-specific transport system (δpstA, δpstB, δpstC, and δpstS).

Additional disruptions that led to a more acidic biolixiviant included those in a hypothetical protein with no similarity to anything previously characterized (δGO_1415); a

gamma-glutamyltranspeptidase (δ*ggt1*); a periplasmic chaperone (δ*surA*); an HTH-type transcriptional regulator (δ*petP*); a two-component system sensor histidine kinase (δ*ykoH*); a pyridoxal 5-phosphate (PLP)-dependent ornithine decarboxylase (δ*speC*); and a TPR domain protein that is a putative component of the TonB iron uptake system (δ*tonB*).

Nine of the tested strains produced biolixiviants significantly higher in pH than pWT (Fig. 4B). The most alkaline biolixiviant was produced by a disruption in the PQQ synthesis system, δ*pqqC*, at pH 4.71. While δ*pqqC* produced very little acid, this is below the pH of glucose in media alone, indicating that some bacterial acidification still occurred. In fact, three of the nine mutants that produce biolixiviants with reduced acidity either synthesize PQQ (δ*pqqC* and δ*tldD*) or use it as a cofactor (δ*mgdh*).

Additional disruptions that led to a more alkaline biolixiviant than pWT include a fructose-bisphosphate aldolase class II (δ*GO_3252*); a GTP and nucleic acid binding protein (δ*ychF*); a lipid A biosynthesis protein (δ*htrB*); a peptide chain release factor (δ*hemK*); the LacI-type transcriptional repressor that increases the initial acidification rate (δ*GO_868*); and the second component of a proteolytic complex with TldD (δ*tldE*).

### Disrupting the phosphate transport system increases bioleaching

We tested whether ten of the mutants that produced a more acidic biolixiviant could bioleach REE from retorted phosphor powder (RPP) from spent fluorescent lightbulbs more efficiently than pWT (Fig. 4C). For each mutant, the elemental composition of REE leachate was similar to that previously reported[26]. Six of these mutants significantly increased bioleaching. Two of the better bioleaching mutants disrupted the *pst* phosphate transport system (δ*pstC* and δ*pstB*). Overall, we found that bioleaching efficiency correlates with biolixiviant pH, as expected (Fig. 4E).

The δ*pstC* mutant produced the most acidic biolixivant and extracted the most REE from RPP: 5.5% total extraction efficiency compared with pWT (4.7%). In other words, δ*pstC* removed 18% more REEs from RPP than pWT. This increase in REE extraction remains significant even after adjusting the α to account for the possibility of a difference due to chance alone (Bonferroni correction, see "Methods" section). Without the adjustment, six of the better acidifiers were also better bioleachers than pWT (Fig. 4C). The remaining better bioleachers increased REE extraction by between 11% (δ*speC*) and 18% (δ*ggt1*) (Fig. 4C).

We speculate that disrupting phosphate transport and signaling derepresses acid production in *G. oxydans*. Six of the disruption strains that resulted in a lower biolixiviant pH (δ*pstC*, δ*pstB*, δ*ggt1*, δ*pstA*, δ*pstS*, and δ*ykoH*), including three that increased bioleaching (δ*pstC*, δ*pstB*, δ*ggt1*), along with many more identified by acidification high-throughput assays (e.g., δ*phoR*, δ*envZ*), are involved in phosphate transport, sensing and signaling.

In its natural environment, *G. oxydans* produces biolixiviants to liberate phosphate from minerals, not metals[38–40]. Under phosphate-limiting conditions, the PstSCAB phosphate transporter will activate the histidine kinase PhoR, which in turn phosphorylates the transcription factor PhoB and activates the *pho* regulon, enabling phosphate solubilization and uptake[41]. Under sufficient phosphate conditions, PhoB is deactivated by PhoR, which in turn inhibits the expression of genes involved in the phosphate-starvation response. We speculate that by disrupting *pstSCAB* or *PhoR*, we prevent *G. oxydans* from sensing when there is adequate phosphate in its environment and when to stop producing biolixiviants.

### Disrupting *mgdh* and PQQ synthesis genes decreases bioleaching

We also tested REE extraction by four mutants that produce a less acidic biolixiviant than pWT, and all were worse bioleachers than pWT (Fig. 4C, D). Unsurprisingly, the δ*mgdh* mutant was the worst bioleacher of all tested mutants, considering its lack of gluconic acid production[35]. δ*mgdh* reduced bioleaching by 97%. Disruption mutants that knocked out the synthesis of mGDH's essential redox cofactor, PQQ, also produced significant reductions in biolixiviant acidity. δ*pqqC* reduced bioleaching by ≈94%. While bioleaching by δ*mgdh* and δ*pqqC* was negligible compared to pWT, they were able to bioleach a statistically significant amount of REE compared to glucose alone. This indicates, as previously speculated[26], that a bioleaching mechanism independent of mGDH exists in *G. oxydans* (Fig. 4D).

Disruption mutants in *tldD* and *tldE* were also much worse at bioleaching than pWT. δ*tldD* reduces bioleaching by 92%, while δ*tldE* reduces it by 63% (Fig. 4C). We speculate that TldD and TldE contribute to the supply of the PQQ cofactor to mGDH. δ*tldD* strongly attenuates acid production (Fig. 4B), and the gene has already been implicated in PQQ synthesis in *G. oxydans* 621H[42]. In *E. coli*, TldD and TldE form a two-component protease for the final cleavage step in the processing of the peptide antibiotic microcin B17[43]. In a similar manner, PqqF and PqqG from *Methylorubrum extorquens* form a protease that rapidly cleaves PqqA, the peptide precursor to PQQ[44]. We speculate that TldD in *G. oxydans* plays the same role as PqqF from *M. extorquens*, while TldE plays the same role as PqqG. Deletion of *pqqF* in *M. extorquens* completely inhibits cleavage of PqqA, while we found that disruption of *tldD* in *G. oxydans* reduces REE bioleaching by 92%. Moreover, deletion of *pqqG* in *M. extorquens* only reduces PqqA cleavage by 50%[44], while disruption of *tldE* only reduces REE bioleaching by 63%. These parallels strongly indicate a role for TldE in the biosynthesis of PQQ in *G. oxydans*.

## Discussion

Bioleaching has the potential to revolutionize the environmental impact of REE production and dramatically increase access to these critical ingredients for sustainable energy technology. However, making REE bioleaching cost competitive with thermochemical methods will require increasing both the rate and completeness (overall efficiency) of REE extraction. This work provides a roadmap for improving bioleaching by genetic engineering.

By constructing a whole genome knockout collection for *G. oxydans*, one of the most promising organisms for REE bioleaching, we are able to characterize the genetics of this process with high sensitivity and high completeness. In total, we identified 165 gene disruption mutants that significantly changed the acidity of its biolixiviant, rate of production, or both. The regulatory elements of each of these genes represent dials that can be turned to improve bioleaching.

As well as producing gluconic acid, *G. oxydans* can be used in the production of industrially important products, including 2,5-diketogluconic acid, a precursor of vitamin C;[45] 5-ketogluconic acid, a precursor of L(+)-tartaric acid used in the production of food and pharmaceuticals[45], vinegar, sorbitol, and dihydroxyacetone[45]. The *G. oxydans* knockout collection will allow characterization of the production of these metabolites and improvement of their production.

REE bioleaching by *G. oxydans* is predominantly controlled by two well-characterized systems: phosphate signaling and incomplete glucose oxidation, which is supported by the production of the redox cofactor PQQ. Interrupting phosphate signaling control of biolixiviant production by disrupting a single gene (*pstC*) can

increase REE extraction by 18%. Disrupting the supply of the PQQ cofactor to the membrane-bound glucose dehydrogenase reduces REE extraction by up to 94%.

Among the many genes whose disruption attenuated final bio-lixiviant acidity, several were related to fructose metabolism. *G. oxydans* converts mannitol (the carbon source used for growth in these studies) into fructose, which can be incorporated into the pentose phosphate and Enter-Douderoff pathways, the latter of which can result in accumulation of acetate in the growth media[46]. As these five disruptions did not affect growth on mannitol, the genes are likely dispensable for fructose catabolism, but may be important for acetate accumulation and some initial media acidification during growth on mannitol. Conversely, disruptions in other genes related to carbohydrate metabolism conferred a faster rate of acidification, most likely due to the disruption of glucose catabolism/incorporation into the pentose phosphate pathway and a preference for incomplete oxidation and gluconic acid production.

Comprehensive screening of the *G. oxydans* genome also revealed targets that could not have been predicted to contribute to REE bioleaching. For example, disrupting *GO_1415*, encoding a protein of unknown function, increases REE bioleaching by 15%. Additionally, our results highlight the potential for a previously uncharacterized role of TldE in PQQ synthesis. Several periplasmic dehydrogenases in *G. oxydans* depend on PQQ for their function[47], and in the case of the D-sorbitol dehydrogenase, mSLDH, overexpression of the *pqq* synthase genes and *tldD* enhances conversion of N-2-hydroxyethyl-glucamine into 6-(N-hydroxyethyl)-amino-6-deoxy-L-sorbofuranose (6NSL), a precursor to the diabetes drug Miglitol[48,49]. The discovery of the potential contribution of TldE to PQQ biosynthesis may allow for exceptional enhancement of cofactor production through the additional overexpression of this gene and a consequent uptick in dehydrogenase activity. PQQ is an essential cofactor important for several other industrial applications of *G. oxydans*, including the production of L-sorbose[50] and 5-keto-D-gluconate[51]. Furthermore, PQQ alone has many applications across many biological processes, from plant protection to neuron regeneration[52].

Our results demonstrate the potential for improving bio-leaching through genetic engineering. Furthermore, the creation of a whole-genome knockout collection in *G. oxydans* will facilitate its use as a model species for further studies in REE bio-leaching and other industrially important applications of similar acetic acid bacteria. The findings of the two major systems contributing to acidification in *G. oxydans* suggest the first steps in the roadmap for greatly improving bioleaching: take the brakes off regulation of acid production by disabling the phosphate-specific transport system, while overexpressing *mgdh* along with the expanded synthesis pathway for its cofactor PQQ.

## Methods

***Gluconobacter oxydans* B58 genome sequencing**. Gluconobacter oxydans strain NRRL B-58 (GoB58) was obtained from the American Type Culture Collection (ATTC), Manassas, VA. In all experiments, GoB58 was cultured in yeast peptone mannitol media [YPM; 5 g L$^{-1}$ yeast extract (C7341, Hardy Diagnostics, Santa Maria, CA), 3 g L$^{-1}$ peptone (211677, BD, Franklin Lakes, NJ), 25 g L$^{-1}$ mannitol (BDH9248, VWR Chemicals, Radnor, PA)] with or without antibiotic, as specified.

Genomic DNA was extracted from saturated culture using a *Quick*-DNA Miniprep kit from Zymo Research (Part number D3024, Irvine, CA). A genomic DNA library was prepared and sequenced using a TruSeq DNA PCR-Free Library Prep Kit (Illumina, San Diego, CA).

The prepared library was sequenced on a MiSeq Nano (Illumina, San Diego, CA, USA) with a 500 bp kit at the Cornell University Institute of Biotechnology (Ithaca, NY, USA). The resulting paired-end reads were trimmed using Trimmomatic[53] and assembled with SPAdes using k-mer sizes 21, 33, 55, 77, 99, and 127 and an auto coverage cutoff[54]. Assembly quality was checked with QUAST[55], and genome completeness was verified with BUSCO[56] using the proteobacteria_odb9 database (available for download at https://datacommons.cyverse.org/browse/iplant/home/shared/iplantcollaborative/example_data/BUSCO.sample.data/proteobacteria_odb9/ancestral) for comparison. The resulting 62 contigs were annotated online using RAST (https://rast.nmpdr.org)[57–59].

**Gene ontology enrichment**. DIAMOND[60] was used to assign annotated protein models with the closest BLAST hit using the Uniref90 database (available for download at https://www.uniprot.org/uniref/), an E-value threshold of 10$^{-10}$, and a block size of 10. InterProScan[61] (version 5.50–84.0) was used to assign family and domain information to protein models.

Output from both of these searches was used to assign gene ontologies with BLAST2GO[62]. Gene set enrichment analysis was performed with the BioConductor topGO package[63] using the default weight algorithm, the TopGO Fisher test, with a *p*-value threshold of 0.05.

**Mating for transposon insertional mutagenesis**. The transposon insertion plasmid pMiniHimarFRT[36] was delivered to GoB58 by conjugation with *E. coli* WM3064. *E. coli* WM3064 transformed with pMiniHimarFRT was grown overnight to saturation in 50 mL LB (10 g L$^{-1}$ tryptone, 5 g L$^{-1}$ yeast extract, and 10 g L$^{-1}$ NaCl) supplemented with 50 µg mL$^{-1}$ kanamycin (kan) and 90 µM diaminopimelic acid (DAP; D1377, Sigma-Aldrich, St. Louis, MO), rinsed once with 50 mL LB, and then resuspended in 20 mL YPM.

GoB58 was grown for approximately 24 h in YPM, diluted to an optical density (OD) of 0.05 in 750 mL YPM and incubated at 30 °C for two doublings until the OD reached 0.2. GoB58 culture was distributed into 13 50 mL conical tubes, to which rinsed and resuspended WM3064 was added at a ratio of 1:1 by density (approximately 1 mL WM3064 to 50 mL B58). Bacteria were mixed by inversion and then centrifuged at 1900×*g* for 5 min. The supernatant was poured off, and the mixture was resuspended in the remaining liquid (≈0.5 mL), dispensed onto a YPM agar petri dish in five spots of 0.1 mL, and allowed to dry on the bench under a flame.

Mating plates were incubated at 30 °C for 24 h. Mating spots were collected by adding 4 mL YPM to a plate, scraping the spots into the liquid, and then suspending bacteria by drawing the liquid up and down several times with a pipette. Suspended cells were collected from each plate, and the suspension was distributed onto petri dishes with YPM agar supplemented with 100 µg mL$^{-1}$ kanamycin at 100 µL per dish.

After 3 days of incubation at 30 °C, colonies were transferred into 96-well microplates using a CP7200 colony picking robot (Norgren Systems, Ronceverte WV, USA). Each well contained 150 µL YPM with 100 µg mL$^{-1}$ kanamycin. For all high-throughput experiments, GoB58 was grown in polypropylene microplates sealed with a sterile porous membrane (Aeraseal, Catalog Number BS-25, Excel Scientific) and incubated at 30 °C with shaking at 800 rpm. Isolated disruption strains were grown for three days to allow cultures in all wells to reach saturated growth. Wells B2 and E7 of each plate were reserved as no-bacteria controls.

A Monte Carlo numerical simulation (collectionmc[36]) was used to approximate how many insertions would need to occur before a mutant is found representing a knockout of each gene in the genome, which demonstrated that approximately 55,000 mutants would need to be generated and selected to identify mutants in at least 99% of all GoB58 genes (Fig. 1B).

In total, 18 matings were required to recover and transfer a progenitor collection of 49,256 disruption strains into 525 microplates over the course of approximately two months. Microplates with saturated cultures were maintained at 4 °C for up to 3 weeks and incubated an additional night at 30 °C before pooling.

The progenitor collection was cataloged using the Knockout Sudoku combinatorial pooling method. Combinatorial pooling was done in three batches. The 525 plates were virtually arranged in a 20 by 27 grid, and creation of the progenitor collection catalog by combinatorial pooling, cryopreservation, pool amplicon library generation, and massively parallel sequencing were all performed using the protocol described by Anzai et al.[36,37].

**Curation of a whole-genome knockout collection**. The condensed *G. oxydans* knockout collection was curated by manual curation of the progenitor collection catalog (as opposed to automatic curation in the Knockout Sudoku protocol[36,37]). To create a condensed collection, a disruption strain was chosen for each of the 2733 disrupted genes available in the progenitor collection, first prioritizing close proximity to the translation start and then the total probability of the proposed progenitor collection address. A second strain was chosen from the remaining strains for each gene that had another available. For 50 genes, the locations for both disruption strains selected were ambiguous, and thus, a third strain was selected from the remaining collection.

In total, 5137 disruption strains were transferred for isolation of single colonies. Many progenitor wells were predicted to have more than one possible strain per well, so for each strain, the number of colonies isolated was two times the predicted number of strains in the progenitor well, up to ten. The condensed collection, which amounted to 17,706 wells, was pooled, sequenced, and validated using the validation procedures detailed in the Knockout Sudoku protocol[36,37]. Unknown disruption strains significantly linked to acidification were identified with Sanger sequencing, also as previously described, with the exception of transposon-specific primers. For the first and second rounds of nested PCR, the transposon-specific primers were (5′-GTATCGCCGCTCCCG-3′, and (5′-CATCGCCTTCTATCGCCTTC-3′), respectively.

**Thymol blue endpoint acidity assay**. Endpoint acidity was measured using the pH indicator thymol blue (TB; SKU114545, Sigma-Aldrich, St. Louis, MO), which changes from red to yellow below a pH of 2.8 (https://www.sigmaaldrich.com/US/

en/product/sial/114545). The lowest pH of biolixiviant generated by GoB58 was 2.3[26]; thus, TB allows for distinguishing strains that lower the pH below that of the wild-type biolixiviant. To generate biolixiviants, the condensed collection was replicated by transferring on pins from cryopreserved culture plates into new growth plates containing 100 μL YPM with 100 μg mL$^{-1}$ kanamycin per well. After two days of growth, an equal volume of 40% w/v glucose was added to saturated cultures for a final solution of 20% w/v glucose. All dispensing into microplates was done with a 96-channel pipette. The amount of glucose needed to lower the pH below 2.3 via the production of gluconic acid was estimated to be 13% w/v, but the higher concentration was used to account for any use of glucose as a carbon source and still maintain an excess amount. Viability tests demonstrated that the bacteria were still viable after two days of culture in such a solution.

Bacteria were incubated with glucose for 48 h to allow acid production to reach completion. Plates were then centrifuged for 3 min at 3200×g (top speed), and 90 μL of the biolixiviant supernatant was removed and added to 10 μL TB at a final concentration of 40 μg mL$^{-1}$. After 1 min of mixing by vortexing, absorbance was measured for each well at 435 and 545 nm on a Synergy 2 plate reader (Biotek Instruments, Winooski, VT, USA). Because of the variation in background absorbance from well to well on each plate, absorbance was measured at these two wavelengths, and their ratio was used as a proxy for pH, which correlates linearly within the range of pH for the majority of biolixiviants produced by the collection (Supplementary Fig. 1).

**Bromophenol blue acidification rate screen.** Acidification rate was measured using the pH-indicating dye bromophenol blue (BPB; 161-0404, Bio-Rad, Richmond, CA). Knockout collection strains were grown for two days. OD was measured at 590 nm for each well, and then 5 μL of culture was transferred to a polystyrene assay plate containing 95 μL of 2% w/v glucose and 20 μg mL$^{-1}$ BPB in deionized water per well. The initial pH of the culture is just above 5, and within moments of adding culture to glucose with BPB, the color begins to change rapidly. Assay plates were mixed by vortexing for one minute after the addition of bacterial culture and then immediately transferred to a plate reader, where the change in color was tracked by measuring absorbance at 600 nm every minute for 6 min, resulting in seven reads. Mean rate (V) and R-squared were calculated by the Gen5 microplate reader and imager software (Biotek Instruments). A plot of all V relative to OD demonstrated that the two are correlated; thus, V was normalized to OD for each well (Supplementary Fig. 3).

**Hit identification in acidification end point and rate screens.** Once every well had its assigned data point (A435/A545 for TB and V/OD for BPB), hits were determined by first identifying outliers for each plate. The interquartile range and upper and lower bounds were calculated in Microsoft Excel considering all wells with cultured disruption strains. Any data point that was more than 1.5× the interquartile range below or above the first and third quartile, respectively, was considered an outlier. A disruption strain was considered a hit if over half of the wells for that strain (or 1 of 2) were outliers.

**End point acidity and rate quantification with colorimetric dyes.** For each assay, knockout strains identified as hits were isolated from the knockout collection into new microplates, along with several blanks per plate, and proxy wild-type strains—GoB58 strains with an intergenic transposon insertion that should not affect the acidification phenotype (see next "Methods" section). Replicates for each knockout strain were distributed across several plates to control for any slight differences between assay plates. For purposes of comparison, all proxy WT strains were considered together; however, OD and acidification phenotypes were measured for each proxy WT strain separately to verify that growth and acidification were unaffected in these strains (Supplementary Fig. 4).

Acidification phenotypes for the disruption strains (for n, see Supplementary Data 6C and D) were compared to that of proxy WT (TB, n = 144; BPB, n = 31) with Student's t-test in Microsoft Excel, two-tailed assuming equal variances. For the acidification rate assay, V/OD data was discarded if the $R^2$ for V was below 0.5. A Bonferroni correction was used to determine significance to account for the possibility a comparison is significant by chance alone: a phenotype was considered significant if p < 0.05/N, where N is the number of comparisons being made (N = 120 or N = 242 for endpoint acidity comparisons with pWT set A or set B, respectively; N = 60 for rate of acidification comparisons with pWT).

**Choice of proxy wild-type comparison.** The biolixiviant end point pH and acidification rate of each G. oxydans mutant were compared against a proxy wild-type set of mutants for each phenotype. To account for the presence of a kanamycin cassette in the genome, the proxy wild-type set for each phenotype was constructed of several mutants with the transposon inserted in an intergenic region that had no growth defect and no apparent change in phenotype (Supplementary Fig. 2).

As the efficiency of E. coli WM3064 to G. oxydans mating was low, construction of the G. oxydans progenitor collection required 18 mating batches. As a result, there were slight variations in the wild-type background from batch to batch, notably two distinct saturated culture densities throughout the collection.

For the acidification rate, these variations did not affect the wild-type behavior across the collection due to normalization with culture density, and a single set of proxy wild-type strains could be used as a comparison with disruption strains in the quantification assays. For the end point pH measurement with TB, these two distinct growth phenotypes were found to affect the result, and thus a proxy wild-type set was used for each. Density measurements across the knockout collection revealed sets of plates resembling each phenotype. Proxy wild-type set A was used as a comparison for insertion mutants located in plates 1–76, 110–130, and 160–185 of the condensed collection (Supplementary Fig. 2A), and proxy wild-type set B, which had a lesser saturated culture density, was used for mutants in plates 77–109 and 130–159 (Supplementary Fig. 2B).

Optical density after 2 days of growth and endpoint acidity using the TB absorbance ratio (A435/A545) were compared for both wild-type sets (Supplementary Fig. 2). For wild-type set A, which was used for the BPB quantification assay, acidification rate of individual proxy WT strains was also compared. Pairwise comparisons were all made using the emmeans package in R with a Tukey p-value adjustment (https://github.com/rvlenth/emmeans).

**Direct measurement of biolixiviant pH.** Bacteria were grown for 48 h in culture tubes containing 4 mL YPM with 100 μg mL$^{-1}$ kanamycin. One tube was left uninoculated as a no-bacteria control. OD was normalized to 1.9 and diluted in half with 40% glucose for a final 20% solution in 1.5 mL. Five replicates were created for each strain and control, and all mixtures were randomly distributed across two 96-well deep well plates. Seven hundred and fifty microliter of mixture was transferred from each well to a second set of deep-well plates for bioleaching experiments. All plates were incubated with shaking at 800 rpm at room temperature.

After two days, one set of deep-well plates was centrifuged for 10 min at 3200×g (top speed), and the pH of the supernatant was measured by insertion of a microprobe to the same depth in each well.

Four standards were used for meter calibration—pH 1, 2, 4, and 7—and the meter was recalibrated after every 12 measurements. pH measurements for each disruption strain (n = 5) were compared with those of proxy WT (n = 15) using Student's t-test in Microsoft Excel, two-tailed, with equal variance. A biolixiviant pH was considered significantly different if p < 0.05/N, with N = 22.

**Direct measurement of REE bioleaching.** The second set of deep-well plates was centrifuged for 10 min at 3200×g (top speed), and 500 μL of biolixiviant was transferred from each well to a 1.7 mL Eppendorf tube. Twenty milligrams (4% w/v) of retorted phosphor powder (gift from Idaho National Lab[26]) was added to each tube for bioleaching. Tubes were shaken horizontally at room temperature and then centrifuged to pellet the remaining solids. Supernatant with leached REE was filtered through a 0.45 μm AcroPrep Advance 96-well filter plate (8029, Pall Corporation, Show Low, AZ, USA) by centrifugation at 1500×g for 5 min.

All samples were diluted 1/200 in 2% trace metal grade nitric acid (JT9368, J.T. Baker, Radnor, PA) and analyzed by an Agilent 7800 ICP-MS for all REE concentrations (m/z: Sc, 45; Y, 89; La, 139; Ce, 140; Pr, 141; Nd, 146; Sm, 147; Eu, 153; Gd, 157; Tb, 159; Dy, 163; Ho, 165; Er, 166; Tm, 169; Yb, 172; and Lu, 175) using a rare earth element mix standard (67349, Sigma-Aldrich, St. Louis, MO) and a rhodium in-line internal standard (SKU04736, Sigma-Aldrich, St. Louis, MO, m/z = 103). Quality control was performed by periodic measurement of standards, blanks, and repeat samples. A pWT biolixiviant sample without bioleaching was spiked with 100 ppb REE standard and analyzed for all REE concentrations as a control. ICP-MS data were analyzed using the program MassHunter, version 4.5.

An additional 1/20 dilution in 2% nitric acid was analyzed for δmgdh and δpqqc disruption strains and the no-bacteria control (glucose).

Bioleaching measurements for each disruption strain (n = 5) were compared with those of proxy WT (n = 15) or glucose (n = 5) using Student's t-test in Microsoft Excel, two-tailed, assuming equal variances. Total REE extracted was considered significantly different if p < 0.05/N, with N = 12 for those compared to pWT and N = 2 for those compared to glucose.

**Reporting summary.** Further information on research design is available in the Nature Research Reporting Summary linked to this article.

## Data availability
Experimental data generated in this study are provided in the Supplementary Information/Source Data file. Raw sequencing data are available through Cornell University eCommons: https://doi.org/10.7298/7s81-5t81. The assembled G. oxydans NRRL B58 genome has been deposited at DDBJ/ENA/GenBank under the accession JAIPVW000000000. The version described in this paper is version JAIPVW010000000.

## Material availability
Correspondence and material requests should be addressed to B.B. Individual strains (up to ≈10 at a time) are available at no charge for academic researchers. We are happy to supply a duplicate of the entire G. oxydans knockout collection to academic researchers but will require reimbursement for materials, supplies and labor costs. Commercial researchers should contact Cornell Technology Licensing for licensing details.

## Code availability

The general purpose Knockout Sudoku software is available at https://doi.org/10.5281/zenodo.5546671. A specific version of the Knockout Sudoku software and input files for creation of the *G. oxydans* knockout collection is available at https://doi.org/10.5281/zenodo.5585726.

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

## Acknowledgements
We thank D. Reed and Y. Fujita at Idaho National Lab for advice and for the gift of retorted phosphor powder. A.M.S. was supported by a Cornell Energy Systems Institute Postdoctoral Fellowship and a Small Grant from the Cornell Atkinson Center for Sustainability. This work was supported by Cornell University startup funds, an Academic Venture Fund award from the Atkinson Center for Sustainability at Cornell University, a Career Award at the Scientific Interface from the Burroughs Welcome Fund to B.B. and by ARPA-E award DE-AR0001341 to B.B., E.G. and M.W.

## Author contributions
Conceptualization, A.M.S. and B.B.; Methodology, A.M.S. and B.B.; Investigation, A.M.S., B.P., S.M. and B.B; Writing—Original draft, A.M.S. and B.B.; Writing—Review and editing, A.M.S., B.P., S.M., M.R., M.W., E.G. and B.B.; Funding acquisition, A.M.S., E.G., M.W. and B.B.; Resources, M.R., E.G. and B.B.; Supervision, M.R. and B.B.; Data curation, A.M.S. and B.B.; Visualization, A.M.S. and B.B.; Formal analysis, A.M.S. and S.M.

## Competing interests
The authors A.M.S, B.P., S.M. and B.B. are pursuing patent protection for engineered organisms using knowledge gathered in this work (US provisional application 63/220,475). The remaining authors declare no competing interests.
