## [Peer Review File · Nature Communications]

Generation of a *Gluconobacter oxydans* knockout collection for improved extraction of rare earth elementsREVIEWER COMMENTS

Reviewer #1 (Remarks to the Author):

The manuscript describes the construction of a single-gene disruption mutant library that was created by knockout sudoku. The library was screened for final acid production and acid production rate as a proxy for bioleaching ability. A set of genes were identified that contribute to either high or low acid production and correlated well with bioleaching ability. The top mutants (those involved in phosphate transport, pqq biosynthesis, and the mgdh) were examined for their ability to extract rare earth elements. Although modest gains were seen, this library represents the starting steps to generate bioleaching engineered strains. Furthermore, this library is certainly useful to investigate the role of other genes and regulatory control in other industrial processes that use *Gluconobacter* sp.

Overall, the manuscript is well written, and the results presented in a clear and convincing way. It was a pleasure to read.

I was a bit confused about the proxy WT set A and set B. What were the 2 behaviors of pWT set A and B. How did they differ? Why was one set chosen over the other for the plates in the end point pH measurement? I would recommend clarify this section in the methods.

I have no major concerns and congratulate the authors on their impressive work.

Reviewer #2 (Remarks to the Author):

A rigorous novel engineering approach (applying the Sudoku knockout method) was used to characterize the metabolism of the microorganism *Gluconobacter* for production of metabolites, such as organic acids, that can lower pH and complex with metals for bioleaching rare earth elements from solid matrix. Furthermore, evaluation of the library was used to identify genes or operons that might be engineered for generating improvements to bioleaching efficiency through increased acidification. Assays were refined to evaluate the library of mutants. And real-world-materials (retorted phosphor powders) containing rare earth elements were tested and compared to the assays. Claims and conclusions are well supported. Overall, the system and analysis suggested success; that is the Sudoku and colorimetric assay approach could be used as a rapid means for engineering microorganisms to identify candidates that might be further engineered to improve metal recovery through bioleaching. Additionally, the use of this knockout system goes far beyond *Gluconobacter* and bioleaching. This system could be used to improve *Gluconobacter* production of metabolites very important for the food and pharmaceutical industry. The system process could also be used with other microorganisms to improve bioleaching or lead to selective recovery or metals. Currently, selective recovery or purification of metals is one of the most costly and environmental impactful processes that lead to reliance on foreign countries modern-day green technology.

An excellent manuscript with very detailed, well described experiments using rigorous controls and analytical methods.

With regards to the rates of acidification experiments could you describe the accuracy and precision (of the change in absorbance of the BPB colorimetric assay) between WT and fast/slow mutants (considering that spectrophotometer value changes might be in the 100th position)?

What accounting of glucose concentration was made in/between assay runs considering that glucose could be used for cell mass or organic acid production/pH decrease? How would a slightly higher/lower concentrations of glucose affect WT/mutant metabolism and the BPB/TB assay results?

The data for figure 3 shows that mannose & fructose metabolism disruption increases pH; are there examples of how these sugars might be involved in production of organic acid and a pH

decrease (i.e., what organic acids or metabolites might be produced upon the sugar catabolism that would lead to a pH decrease with the WT). The disruption of these pathways also increased the acidification rate (why would their metabolism slow the acidification, perhaps directing resources away for glucose conversion to gluconic acid).

Please remove jargon, e.g., "spun down" (centrifuged), "struck out for single colonies" (transferred for isolation), "pipetted" (dispensed), etc.

21st September 2021

To Reviewers 1 and 2,

We have prepared a revised manuscript (“*Gluconobacter oxydans Knockout Collection Finds Improved Rare Earth Element Extraction*”, Schmitz *et al.*) in response to your comments. We are very grateful for these thoughtful and constructive comments. We have sought to address all comments and questions either in a revised copy of the manuscript or in this letter.

Text from the reviews are quoted in *italics*. The location of changes to the manuscript are noted in responses to specific comments by the reviewers and are highlighted in red or blue (two colors are used to distinguish separate consecutive changes) in the manuscript itself along with a tag that denotes the reviewer comment this change addresses (e.g., **RC1.1**).

To find all changes associated with a reviewer comment, search in the main text and supplementary information for these tags (e.g., **RC1.1**). All changes requested by the editorial team are denoted by **E** (e.g., **E1**).

To help track the reviewer comments and corresponding responses, we have summarized the changes and their location in the revised manuscript in a table attached to the end of this letter.

Response to Reviewer 1

Reviewer 1 comments:

“The manuscript describes the construction of a single-gene disruption mutant library that was created by knockout sudoku. The library was screened for final acid production and acid production rate as a proxy for bioleaching ability. A set of genes were identified that contribute to either high or low acid production and correlated well with bioleaching ability. The top mutants (those involved in phosphate transport, pqq biosynthesis, and the mgdh) were examined for their ability to extract rare earth elements. Although modest gains were seen, this library represents the starting steps to generate bioleaching engineered strains. Furthermore, this library is certainly useful to investigate the role of other genes and regulatory control in other industrial processes that use Gluconobacter sp.”

“Overall, the manuscript is well written, and the results presented in a clear and convincing way. It was a pleasure to read.”

Specifically, **Reviewer 1** comments:

RC1.1 *“I was a bit confused about the proxy WT set A and set B. What were the 2 behaviors of pWT set A and B. How did they differ? Why was one set chosen over the other for the plates in the end point pH measurement? I would recommend clarify this section in the methods.”*

We really appreciate this comment from **Reviewer 1**: many thanks for pointing out this confusion. We agree that the different proxy WT set phenotypes were not described well. The two distinct phenotypes arose in the saturated densities of the cultures. Proxy WT set B had a lower saturated culture density than set A. This was evident in the density measurements of the entire collection, and specific batches of plates

could be identified to correspond with each phenotype. We have updated the Methods section of the main text to better explain what was observed and what was done as a result.

Finally,

“I have no major concerns and congratulate the authors on their impressive work.”

Response to Reviewer 2

Reviewer 2 comments:

*“A rigorous novel engineering approach (applying the Sudoku knockout method) was used to characterize the metabolism of the microorganism *Gluconobacter* for production of metabolites, such as organic acids, that can lower pH and complex with metals for bioleaching rare earth elements from solid matrix. Furthermore, evaluation of the library was used to identify genes or operons that might be engineered for generating improvements to bioleaching efficiency through increased acidification. Assays were refined to evaluate the library of mutants. And real-world-materials (retorted phosphor powders) containing rare earth elements were tested and compared to the assays. Claims and conclusions are well supported. Overall, the system and analysis suggested success; that is the Sudoku and colorimetric assay approach could be used as a rapid means for engineering microorganisms to identify candidates that might be further engineered to improve metal recovery through bioleaching. Additionally, the use of this knockout system goes far beyond *Gluconobacter* and bioleaching. This system could be used to improve *Gluconobacter* production of metabolites very important for the food and pharmaceutical industry. The system process could also be used with other microorganisms to improve bioleaching or lead to selective recovery of metals. Currently, selective recovery or purification of metals is one of the most costly and environmental impactful processes that lead to reliance on foreign countries modern-day green technology.”*

“An excellent manuscript with very detailed, well described experiments using rigorous controls and analytical methods.”

Reviewer 2 makes four specific recommendations for revisions that we address here and in the revised manuscript.

RC2.1 *“With regards to the rates of acidification experiments could you describe the accuracy and precision (of the change in absorbance of the BPB colorimetric assay) between WT and fast/slow mutants (considering that spectrophotometer value changes might be in the 100th position)?”*

Indeed the changes in absorbance were small, but we were able to use the R^2 for V generated by the Gen5 software as a measure of accuracy, and discarded any rate measurements with $R^2 < 0.5$. We realize that this was not included in the methods and have added it to the main text. Furthermore, the R^2 for each V calculation is included in the data table available with this submission. As for precision, panels D and E in **Figure 2** have been updated to include error bars and individual data points.

RC2.2 *“What accounting of glucose concentration was made in/between assay runs considering that glucose could be used for cell mass or organic acid production/pH decrease? How would a slightly higher/lower concentrations of glucose affect WT/mutant metabolism and the BPB/TB assay results?”*

Great question - while we cannot rule out the possibility that there were slight variations in glucose concentration from plate to plate, we controlled for their effect by distributing the strain replicates among several plates. Mixing of bacteria with glucose was done simultaneously across all wells of each plate

using a recently calibrated 96-channel pipette. We realize the methods did not make this clear, and have updated the text to reflect the additional detail. Furthermore, panels D and E in Figure 2 were missing error bars and individual data points, which we have added, and we have updated the corresponding figure caption.

RC2.3 *“The data for figure 3 shows that mannose & fructose metabolism disruption increases pH; are there examples of how these sugars might be involved in production of organic acid and a pH decrease (i.e., what organic acids or metabolites might be produced upon the sugar catabolism that would lead to a pH decrease with the WT). The disruption of these pathways also increased the acidification rate (why would their metabolism slow the acidification, perhaps directing resources away for glucose conversion to gluconic acid).”*

Thank you for drawing our attention to this. While we can only speculate, further reading allows us to predict that the higher final pH upon disruption of fructose metabolism may be due to the loss of acetate accumulation in the media, which has been shown for growth on mannitol. Conversely, we also speculate that disruption of other genes involved in carbohydrate metabolism would limit glucose metabolism and redirect it towards conversion to gluconic acid, as suggested. We have clarified this observation in the results section and elaborated on its significance in the conclusions, including the addition of a reference.

RC2.4 *“Please remove jargon, e.g., “spun down” (centrifuged), “struck out for single colonies” (transferred for isolation), “pipetted” (dispensed), etc.”*

We really appreciate this suggestion. We have updated the text throughout the methods section.

We would like to thank the reviewers for their comments which have allowed us to improve this article. We hope the revised manuscript is acceptable.

Response Summary

Comment Number	Comment	Response Summary	Response Start Location(s)
RC1.1	I was a bit confused about the proxy WT set A and set B. What were the 2 behaviors of pWT set A and B. How did they differ? Why was one set chosen over the other for the plates in the end point pH measurement? I would recommend clarify this section in the methods.	Updated methods section to clarify what was observed and done.	Main Text Lines 394-402
RC2.1	With regards to the rates of acidification experiments could you describe the accuracy and precision (of the change in absorbance of the BPB colorimetric assay) between WT and fast/slow mutants (considering that spectrophotometer value changes might be in the 100th position)?	Updated methods section. Revised Figure 2D and 2E to include all data points and error bars.	Main Text Line 381
RC2.2	What accounting of glucose concentration was made in/between assay runs considering that glucose could be used for cell mass or organic acid production/pH decrease? How would a slightly higher/lower concentrations of glucose affect WT/mutant metabolism and the BPB/TB assay results?	Updated methods section. Revised Figure 2D and 2E to include all data points and error bars.	Main Text Lines 375-377
RC2.3	The data for figure 3 shows that mannose & fructose metabolism disruption increases pH; are there examples of how these sugars might be involved in production of organic acid and a pH decrease (i.e., what organic acids or metabolites might be produced upon the sugar catabolism that would lead to a pH decrease with the WT). The disruption of these pathways also increased the acidification rate (why would their metabolism slow the acidification, perhaps directing resources away for glucose conversion to gluconic acid).	Added text to both results and discussion sections.	Main Text Lines 130-134, 136, 230-238
RC2.4	Please remove jargon, e.g., “spun down” (centrifuged), “struck out for single colonies” (transferred for isolation), “pipetted” (dispensed), etc.	Main text (mostly methods) has been edited lightly throughout to eliminate jargon.	Main Text Lines 84, 287-412
E1	Editorial policy checklist: https://www.nature.com/documents/nr-editorial-policy-checklist.pdf	Completed. Please see attached document.	
E2	Reporting summary: https://www.nature.com/documents/nr-reporting-summary.pdf	Completed. Please see attached document.	
E3	* If your paper uses custom code/software, please complete the following code and software submission checklist and make your code available for reviewer assessment, if you have not already done so. The code/software can be provided in a zip file with a readme.txt file or other instructions for installing and running the software. If appropriate, also provide example data and expected output. If you have any issues with the file upload, please let me know. https://www.nature.com/documents/nr-software-policy.pdf	Completed. Please see attached document.	
E4	* All Nature Communications manuscripts must include a “Data Availability” section after the Methods section but before the References. If any of the data can only be shared on request or are subject to restrictions, please specify the reasons and explain how, when, and by whom the data can be accessed. For more information on this policy and a list of examples, see: https://www.nature.com/documents/nr-data-availability-statements-data-citations.pdf	Please see an updated Data Availability section in the revised main text.	Main Text Lines 445-446
E5	* If your paper uses custom code/software, please also include a “Code Availability” section after the “Data Availability” section. If the code can only be shared on request, please specify the reasons. For more information on our code sharing policy and requirements, please see: https://www.nature.com/nature-research/editorial-policies/reporting-standards#availability-of-computer-code	This statement was included in the original submission.	Main Text Line 448

Comment Number	Comment	Response Summary	Response Start Location(s)
E6	* All novel microarray, DNA sequencing, RNA-seq or proteomic datasets must be deposited in a publicly accessible database, and accession codes and associated hyperlinks must be provided in the "Data Availability" section.	This Whole Genome Shotgun project has been deposited at DDBJ/ENA/GenBank under the accession JAIPVW000000000. The version described in this paper is version JAIPVW010000000.	Main Text Lines 446-448
E7	* We strongly encourage you to deposit all new data associated with the paper in a persistent repository where they can be freely and enduringly accessed. We recommend submitting the data to discipline-specific and community-recognized repositories; a list of repositories is provided here: http://www.nature.com/sdata/policies/repositories Refer to our data policies here: https://www.nature.com/nature-research/editorial-policies/reporting-standards#availability-of-data	All raw sequencing data has been deposited to Cornell University eCommons at https://doi.org/10.7298/7s81-5t81	Main Text Lines 445-446
E8	To maximise the reproducibility of research data, we strongly encourage you to provide a file containing the raw data underlying the following types of display items: - Any reported means/averages in box plots, bar charts, and tables - Dot plots/scatter plots, especially when there are overlapping points - Line graphs The data should be provided in a single Excel file with data for each figure/table in a separate sheet, or in multiple labelled files within a zipped folder. Name this file or folder 'Source Data', and include a brief description in your cover letter. The "Data Availability" section should also include the statement "Source data are provided with this paper." To learn more about our motivation behind this policy, please see: https://www.nature.com/articles/s41467-018-06012-8	Please see attached zip archive with all source data included with the resubmission.	Main Text Lines 445-446
E9	* We also mandate the presentation of uncropped versions of any gels or blots, labelled with the relevant panel and identifying information such as the antibody used.	N/A. No gels or blots are included in this work.	
E10	* Please replace any bar graphs with plots that feature information about the distribution of the underlying data. All data points should be shown for plots with a sample size less than 10. For larger sample sizes, please consider box-and-whisker or violin plots as alternatives. Measures of centrality, dispersion and/or error bars should be plotted and described in the figure legend.	Figures 2D and E, and all panels of Figure 4 have been updated with all data points shown. Captions have been updated accordingly.	Figures 2D, 2E, and 4
E11	* Nature Communications is committed to improving transparency in authorship. As part of our efforts in this direction, we are now requesting that all authors identified as 'corresponding author' create and link their Open Researcher and Contributor Identifier (ORCID) with their account on the Manuscript Tracking System prior to acceptance. ORCID helps the scientific community achieve unambiguous attribution of all scholarly contributions.	Included. All authors inputted their ORCIDs during submission.	
E12	Please also see the Nature Communications formatting instructions , which you may find useful while preparing your revised manuscript.	Shortened abstract to fewer than 150 words. Headings were renamed. Subheadings were shortened to fewer than 60 characters (without spaces). Secondary subheadings were removed.	Please see all points in the Main Text marked E12

REVIEWERS' COMMENTS

Reviewer #1 (Remarks to the Author):

The authors have addressed all comments. I really enjoyed reading it.

Reviewer #2 (Remarks to the Author):

Thank you for your thoughtful responses. I recommend publication. Congratulations!